# Experience of switching from a daily to a less frequent administration of injection treatments

Jane Loftus[1]*, Andrew Yaworsky[2], Carl L. Roland[3], Diane Turner-Bowker[2], Megan McLafferty[2], Sylvia Su[2], Roger E. Lamoureux[2]

1 Pfizer, Ltd, United Kingdom, 2 Adelphi Values, Boston, MA, United States of America, 3 Pfizer Inc, Sanford, NC, United States of America

* Jane.Loftus@Pfizer.com

**Data Availability Statement:** All relevant data are within the paper and its Supporting information files.

## Abstract

### Background

Daily injections of recombinant human growth hormone are the standard of care to treat growth failure due to pediatric growth hormone deficiency (GHD). While effective, daily injections are burdensome and can compromise adherence. In recent years, novel injection treatments requiring less frequent administration for growth hormone deficiency (GHD) have been developed. A targeted, pragmatic literature review was conducted to summarize and document the patient experience of moving from daily to less frequent injections, with a specific focus on changing from daily to weekly injection treatments in pediatric GHD (pGHD).

### Objective

Explore and describe the patient experience when switching from a daily to a less frequent injection schedule for GHD.

### Methods

Targeted literature searches were conducted to identify literature describing the patient experience of moving from a daily to weekly injection in GHD. Supplementary searches were conducted to identify literature describing the patient experience of moving from daily to less frequent injection regimens in other medical conditions.

### Results

Across searches, 1,691 abstracts were reviewed and 13 articles were included in the final analysis. These publications reported that patients moving to less frequent injections across a variety of conditions, including GHD, experienced increased convenience and satisfaction, higher adherence rates, fewer adverse events, and improved quality of life. Less frequent injections were also reported to be at least as efficacious as daily treatments.

**Funding:** The study and research reported in this manuscript was sponsored by Pfizer (https://www.pfizer.com/). Representatives from Pfizer participated in conceptualizing the study design, the decision to publish results, and preparing the manuscript. JL conceptualized the study design. JL and CR informed the decision to publish the results and participated in preparation of the manuscript. Pfizer's publications team reviewed and approved this manuscript without edits.

**Competing interests:** JL and CLR are full-time employees of Pfizer, who funded the study. AY, MM, SS, and REL are employees of Adelphi Values LLC, who were paid consultants to Pfizer in connection with this research and development of this manuscript. DTB was an employee of Adelphi Values LLC at the time that the research was conducted. This does not alter co-author adherence to PLOS ONE policies on sharing data and materials.

## Conclusions

Less frequent injections in GHD and as other conditions are less burdensome, positively benefit patients, and result in improved adherence that may lead to improved clinical outcomes. Clinicians may consider weekly regimens as an effective alternative for patients, in particular in pGHD, especially when missed injections can negatively impact treatment outcomes. More research is needed to better understand the real-world benefits of injectable therapies that require less frequent administration (e.g., weekly versus daily).

## Introduction

The prevalence of pediatric growth hormone deficiency (pGHD) is estimated at 1:4,000 to 1:10,000 [1–4]. The most apparent feature of pGHD is growth failure or restriction that can lead to short stature, which has been linked to decreased quality of life (QoL) [5–7]. GHD is also associated with metabolic consequences (including impaired lipid metabolism, protein synthesis, and bone mineralization).

The current standard of care is recombinant human growth hormone (r-hGH) which has been used for several decades to treat adults and children with GHD. In children, r-hGH therapy aims to: (i) increase height during childhood, (ii) attain adult height targets, (iii) minimize adverse events (AEs), (iv) achieve cost-effective treatment [8], and (v) improve QoL [9]. In addition, in adults, treatment aims with r-hGH include improving QoL and reducing cardio- and cerebrovascular morbidity.

In pGHD, r-hGH therapy is administered daily via subcutaneous (SC) injections given over an extended timeframe (e.g., five or more years), until the patient's final height target has been attained. Given the expected long-term duration of daily r-hGH treatments, optimal treatment compliance is a challenge, which could be attributed to the burden of daily injection regimens [10, 11]. A published web-based survey of 61 patients (age ≥13) treated with daily r-hGH and 239 caregivers found that 62.3% of patients and 51% of caregivers reported missing at least one dose each month. The most frequently endorsed reason for missing a dose was being away from home or traveling (12.3%) and special events (10.1%) [12]. Non-compliance has been shown to be the most common cause of reduced height velocity in children [11, 13, 14]. Studies have also demonstrated a reduction in compliance over time [14].

New therapies for GHD designed to be administered by weekly SC injection are in development as an alternative treatment to the current standard of care. It is hypothesized (similar to what has been demonstrated in oral treatment regimens when comparing daily and weekly administration for other conditions [15–17]) that changing from daily to weekly injections for pGHD will result in improved compliance/adherence, and, ultimately, clinical outcomes (e.g., growth velocity and final height in pGHD) [2–6].

In its recently issued draft guidance on patient-focused drug development, the United States Food and Drug Administration highlighted the importance of incorporating patient experience in the treatment development process (including patient experience with and preference for treatments, and the relative importance of outcomes) [18]. In line with this guidance, the authors explored and documented the patient experience when switching from a daily treatment regimen to a less frequent regimen as reported in empirical literature.

This paper presents the methods used to conduct a targeted, pragmatic review of empirical literature to identify information related to the potential benefits that patients may experience when switching from a daily to a weekly injection regimen for GHD. Supplementary searches

were also conducted to identify and describe the patient experience of a switch from daily to other less frequent injection regimens (e.g., once every two weeks, once a month) in GHD and other conditions. The results from this research can inform our understanding of the patient experience of a less frequent injection regimen, particularly in GHD.

## Objective

The primary objective of this research was to explore and document the experiences of patients when switching from a daily to a weekly injection regimen to treat GHD. A secondary objective was to identify and describe the patient experience of a switch from a daily to other less frequent injection regimens in conditions other than GHD.

## Materials and methods

### Initial targeted search: Patient experience of daily and weekly injection regimens in GHD

A pragmatic and targeted search of empirical literature was conducted. A search strategy (Table 1) was developed to identify abstracts of interest using search terms related to human growth hormone treatments (including names of generic and brand name treatments), injection regimen and frequency, and the impact of treatment (e.g., convenience, burden, emotional functioning, activities of daily living). The search strategy also included terms related to following a prescribed injection regimen (compliance and adherence). It should be noted that "adherence" and "compliance" in the reviewed publications are often used interchangeably and in some cases are considered synonymous [19]. For ease of reading and consistency, the authors have used "adherence" to denote patients following a prescribed treatment regimen as directed throughout this paper. A PICOS framework (Fig 2) was utilized, and articles selected

**Table 1. Search terms applied in OvidSP to identify potential publications for review.**

| Search and focus | Categories and search terms[a] | | | | |
|---|---|---|---|---|---|
| | **Drug name treatments for GHD** | **Other GHD treatments** | **Injection** | **Frequency** | **Impact of treatment** |
| **Primary search**[b] | "Norditropin", "Nutropin AQ", "Omnitrope", "Saizen", "Zomacton", "Humatrope", "Serostim", "Nutropin Depot", "Accretropin", "Genotropin", Norditropin FlexPro", "Omnitrope", "Declage", "Jintrolong", "TransCon Growth Hormone", "Somapacitan", "HyTropin", "MOD-4023 | "somatotropin", "human growth hormone", "recombinant human growth hormone", "tesamorelin", "sermorelin" | "injection", "administration", "regimen", "adherence", "compliance", persistence" | "frequency", "daily", "weekly", "two weeks", "biweekly", "monthly" | "convenience", "impact", "complaint", "side effect", "burden", "quality of life", "preference", "value", "psychological", "psychosocial", "value", "activities of daily living", "everyday life", "social", "emotional" |
| **Additional search 1**[c] | N/A | N/A | | "frequency", "daily", "weekly" | |
| **Additional search 2**[d] | N/A | N/A | | "frequency", "daily", "weekly", "two weeks", "biweekly", "monthly", "bimonthly" | |

[a] This presents a broad overview of the search terms used and is not an exhaustive list

[b] The search that was run in the in the MEDLINE®, Embase, and PsycINFO® databases using the OvidSP platform

[c] A supplemental search to explore the patient experience of switching from a daily injection regimen to a weekly injection regimen in conditions other than GHD

[d] A supplemental search to explore the patient experience of switching to other less frequent injection regimens (e.g., monthly) in conditions other than GHD

Note: GHD = growth hormone deficiency

for potential full-text inclusion (for the initial search and all subsequent searches) were compared to the criteria in this framework, before being included in the final analysis.

The search was run in the MEDLINE®, Embase, and PsycINFO® databases using the OvidSP platform. Given the recent availability of non-daily human growth hormone treatments, only abstracts presenting data from studies published within 10 years of the conducted search on 26 February 2020 were considered for inclusion. The search was also limited to English-language research studies in humans. The resulting abstracts were screened using Abstrackr [20], a web-based program that allows researchers to review abstracts and document whether or not each abstract was relevant to the research question. Abstracts were considered for inclusion if they included results related to GHD in patients that had moved from a once daily to a once weekly injection regimen.

Reference lists of selected articles were also reviewed, and supplementary grey literature searches were conducted in Google Scholar. Publications were excluded from full review if they primarily focused on efficacy or clinical findings (rather than the patient experience) or did not report on outcomes due to a change in injection regimen. Researchers read each article to identify relevant content. Information from each study was documented to characterize the study (e.g., study design, patient sample, evaluated treatment[s], administration regimens) and summarize study findings and conclusions as they relate to the patient experience with adherence, treatment-burden, and QoL impacts when switching from a daily to a weekly injection regimen. Studies selected for full-text review were evaluated against the population, intervention, comparison, outcome, and study type (PICOS) framework [21–23] to ensure that each study met the defined criteria.

### Additional searches—Patient experience of less frequent injection regimens in other conditions

As the initial search identified a paucity of research in the target area, two additional targeted searches (Table 1) were conducted using the same methodology previously described to explore the patient experience of switching from daily to less frequent injection regimens in conditions other than GHD. Given the magnitude of search results and practical limitations, a pragmatic approach was used during abstract screening. Understanding that OvidSP orders search results by displaying the most relevant first (based on the number of search terms contained in the abstract), the first 300 abstracts were initially screened. During screening, the research team considered the number of relevant publications being identified, and the unique additional information that each might provide to help answer the research questions. If a large number of new and relevant publications were still being identified at the end of screening the first 300 abstracts, screening of additional abstracts beyond the initial 300 would be considered.

The first additional search, conducted on 18 March 2020, sought information on the patient experience of switching from a daily to a weekly injection regimen in conditions other than GHD. The second additional search, conducted on 14 April 2020, sought information on the patient experience of switching to other less frequent injection regimens (e.g., monthly) in conditions other than GHD.

## Results

### Search results and publication selection

**Initial targeted search—Patient experience of daily and weekly injection regimens in GHD.** The initial targeted search yielded 255 abstracts. All abstracts were reviewed, and eight

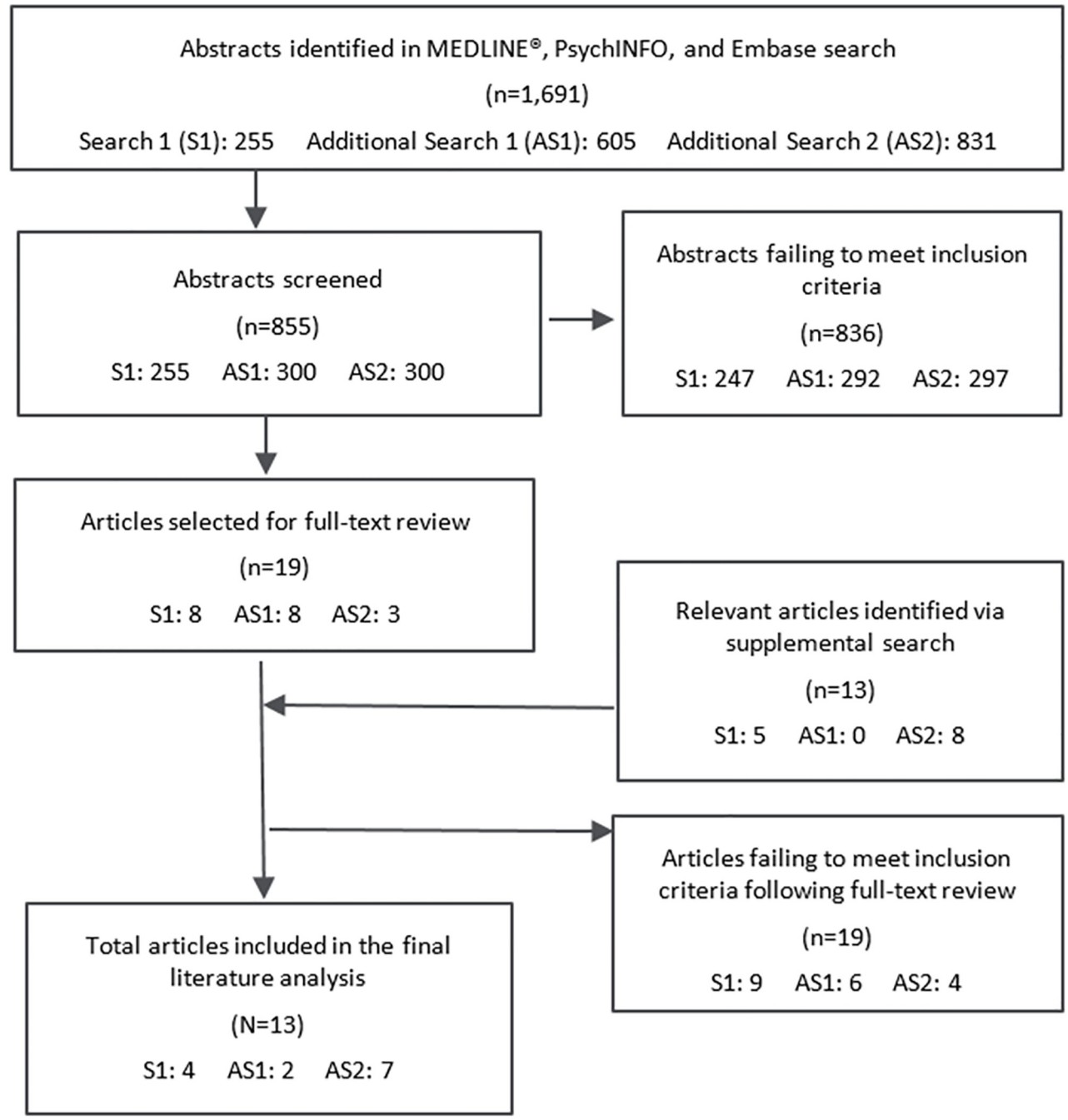

**Fig 1. Search flow diagram for the selection of publications reviewed.** Note: S1 = Search 1; AS1 = Additional Search 1; AS2 = Additional Search 2.

publications were initially selected for full-text review (Fig 1). After review, six of these publications were excluded because they did not directly compare injection regimens (n = 4) or they reported on the same study data as another publication already reviewed (n = 2). Two additional publications, identified via Google Scholar search and reference lists of reviewed publications, were also included for full review. As a result, four publications presenting information related to the patient experience switching from daily to weekly injection

- **Patient problem (or population):** Patients with any disease type receiving oral or injection treatment on a consistent treatment regimen (e.g., daily, weekly)
- **Intervention:** Oral or injection treatment of any type administered on a less frequent consistent regimen
- **Comparison or Control:** 'Comparison' between patients' experience of the original treatment regimen and the less frequent regimen as assessed by the outcomes listed below
- **Outcome:** The following outcomes were of interest in this research study: Adherence, Burden, Compliance, Convenience, Health-related quality of life, Preference, Safety, or Satisfaction
- **Study Type:** Regulated clinical trials or observational studies

| Reference [manuscript ref. #] | P | I | C | O | S |
|---|---|---|---|---|---|
| Johannsson 2020 [21] | ✓ | ✓ | ✓ | ✓ | ✓ |
| Johannsson 2018 [22] | ✓ | ✓ | ✓ | ✓ | ✓ |
| McNamara 2020 [23] | ✓ | ✓ | ✓ | ✓ | ✓ |
| Humphriss 2017 [24] | ✓ | ✓ | ✓ | ✓ | ✓ |
| Qiao 2016 [16] | ✓ | ✓ | ✓ | ✓ | ✓ |
| Hauber 2016 [25] | ✓ | ✓ | ✓ | ✓ | ✓ |
| Zuurbier 2016 [28] | ✓ | ✓ | ✓ | ✓ | ✓ |
| Veneziano 2017 [26] | ✓ | ✓ | ✓ | ✓ | ✓ |
| Cutter 2020 [27] | ✓ | ✓ | ✓ | ✓ | ✓ |
| Osborne 2012 [30] | ✓ | ✓ | ✓ | ✓ | ✓ |
| Mathews 2019 [31] | ✓ | ✓ | ✓ | ✓ | ✓ |
| Cornford 2018 [32] | ✓ | ✓ | ✓ | ✓ | ✓ |
| Khan 2012 [29] | ✓ | ✓ | ✓ | ✓ | ✓ |

**Fig 2. PICOS framework for targeted literature review.** Note: P = Patient problem (or population); I = Intervention; C = Comparison or control; O = Outcome; S = Study type.

regimens in GHD were included in the final analysis. All articles included in the final analysis, from the initial search and subsequent searches, met the criteria outlined in the PICOS framework (Fig 2).

**Additional search 1: Patient experience of daily and weekly injection regimens in any condition (other than GHD).** The first additional literature search, designed to describe the patient experience of switching from daily to weekly injection regimens in conditions besides GHD, identified 609 abstracts (Fig 1). The first 300 abstracts were screened; a downward trend

in the number of new and relevant publications was observed as screening proceeded. In accordance with the described methodology, screening was halted, and eight publications were selected for full review. Of these, six were excluded from analysis, as they did not directly compare injection regimens. As a result, two articles from this search were included in the final analysis.

**Additional search 2: Patient experience of daily and other less frequent injection regimens in any condition (other than GHD).** The second additional literature search, designed to describe the patient experience of switching from daily to other less frequent (e.g., monthly) regimens among patients in non-GHD conditions, identified 831 abstracts (Fig 1). The first 300 abstracts were screened. A downward trend in new and relevant publications was observed as screening proceeded; therefore, screening was halted and three publications were selected for full review. Additional searches conducted using Google Scholar and the reference lists of reviewed publications identified eight publications for full-text review. Of the 11 publications were reviewed in full, four were excluded from analysis as they did not directly compare injection regimens. As a result, seven publications from this search were included in the final analysis.

### Data extraction and analysis

A total of 13 publications were included in the final analysis (four from the primary search, two from the first additional search, and seven from the second additional search), summarizing data across five therapeutic areas, clinical trial designs, and injection regimens (Table 2).

**Initial targeted search: Patient experience of daily and weekly injection regimens in GHD.** The four publications reviewed from the initial search reported information related to several different growth hormone (GH) treatments, including once-weekly somapacitan and once-daily GH replacement therapy (Table 2).

Two publications reported on adherence in clinical trials which, as expected, was high in both the once weekly and daily treatment groups [24, 25]. Johannsson et al. (2020) reported the mean adherence for weekly aGHD treatment to be 95.5% compared to 90.6% for daily aGHD treatment [24]. In addition, it was found that the weekly (72.5% of patients) injection regimen resulted in the same or fewer AEs during the main study period when compared to the daily (79.8% of patients) injection regimen [24, 25]. Further, efficacy was equivalent between daily and weekly injection regimens over the course of the study [24]. Johannsson et al. (2018) found a statistically significant difference (p = 0.0171) in convenience score (8.22 from randomization to Week 26 favoring weekly somapacitan) on the Treatment Satisfaction Questionnaire for Medication-9, which demonstrated that weekly injections were more convenient than daily injections to patients [25].

McNamara et al. (2020) conducted a discrete choice experiment (DCE), in which pediatric, adolescent, and adult patients with GHD were provided with identical injection configurations that differed only by frequency of injections. Findings demonstrated that patients across all age groups preferred less frequent injections to treat GHD; specifically, 76%, 75%, and 70% of participants indicated they would choose a once-monthly, once-weekly, or a once-every-two-weeks injection regimen over their current daily regimen, respectively [26].

A long-term safety study conducted by Humphriss et al. (2017) involved pre-pubescent patients receiving twice-monthly injections with GHD over a 24-month period. Although this publication was not strictly within the focus of the search, it directly compared injection regimens in GHD and provides additional information on the patient experience of less frequent injections in that condition. Specifically, in the initial Phase 2 portion of the study, participants were randomized to once weekly, twice monthly, and once monthly regimens for six months.

**Table 2. Summary of studies included in final review.**

| Reference (et al.) | Therapeutic area | Treatment(s) and/or regimens studied | Type of study/study design | Principal outcome (e.g., adherence, AE, efficacy) |
|---|---|---|---|---|
| Johannsson 2020 [24] | GHD (adult) | Once-weekly somapacitan vs. once-daily growth hormone replacement therapy | Regulated clinical trial | Compliance and safety |
| Johannsson 2018 [25] | GHD (adult) | Once-weekly somapacitan vs. once-daily Norditropin® | Regulated clinical trial | Compliance, satisfaction, convenience, and safety |
| McNamara 2020 [26] | GHD (pediatric, adolescent, adult) | Once-daily vs. once-weekly vs. once every two weeks vs. once monthly r-hGH injections | Observational study (DCE) | Preference |
| Humphriss 2017 [27] | GHD (pediatric) | Weekly vs. twice-weekly vs. monthly Somavaratan | Regulated clinical trial | Compliance |
| Qiao 2016 [16] | Type 2 diabetes | Once-weekly Exenatide (Bydureon®) vs. once-daily Liraglutide (Victoza®) | Observational study (longitudinal database review) | Adherence |
| Hauber 2016 [28] | Type 2 diabetes | Once-weekly Exenatide vs. once-daily liraglutide vs. insulin vs. no injectable | Observational study (DCE) | Preference |
| Zuurbier 2016 [29] | MS | Three-times-weekly vs. daily GA | Observational study (online survey) | Adherence |
| Veneziano 2017 [30] | MS | Three-times-weekly vs. once daily GA (Copaxone®) | Regulated clinical trial | Satisfaction, convenience, safety, and adherence |
| Cutter 2020 [31] | Relapsing-remitting MS | Three-times-weekly vs. daily GA | Regulated clinical trial | Satisfaction, convenience, health-related quality of life (HRQoL) |
| Osborne 2012 [32] | Schizophrenia | Once every two weeks Risperidone vs. once every two or four weeks (depending on dose) Olanzapine vs. once every three months Paliperidone palmitate | Observational study (time trade-off vignette) | HRQoL |
| Mathews 2019 [33] | Schizophrenia | Once monthly vs. Once every three months Paliperidone palmitate | Regulated clinical trial | Convenience and adherence |
| Cornford 2018 [34] | Prostate cancer | Once every six months Triptorelin vs. once monthly or once every three months luteinizing hormone-releasing hormone agonist | Observational study (real world retrospective) | Treatment burden, HRQoL |
| Khan 2012 [35] | Prostate cancer | Once every 12 months Vantas® vs. once every six months Decapeptyl® vs. once every three months or once monthly hypothetical treatments | Observational study (patient survey) | Preference and convenience |

Note: AE = adverse event, DCE = discrete choice experiment, GA = glatiramer acetate, GHD = growth hormone deficiency, HRQoL = health-related quality of life, MS = multiple sclerosis, r-hGH = recombinant human growth hormone

All participants were then transitioned to the twice monthly dose for the long-term safety study. Results demonstrated that dosing adherence was 99.6% in patients on the twice-monthly injection regimen over the 24-month observation period during the long-term safety study [27]. The authors indicate that a less frequent injection schedule may improve treatment adherence.

**Additional search 1: Patient experience of daily and weekly injection regimens in any condition (other than GHD).** From the search conducted on 18 March 2020, two publications were reviewed relating to patient experience switching from daily to weekly injection regimens in type 2 diabetes.

One publication that used a nationwide longitudinal prescription database to analyze adherence to therapy [16] found that adherence, calculated as proportion of days covered (PDC—calculated as total number of days supplied per patient during the six-month period divided by 180 days) by prescription, was significantly higher for type 2 diabetes patients taking once-weekly injections than those taking once-daily injections. The PDC for once weekly injections was 0.88, and 0.77 for once daily injections ($p < 0.05$). The other publication involved

a DCE [28], where among a sample of type 2 diabetes patients, preference for weekly injections over daily injections was the most important attribute reported by patients across the sample.

**Additional search 2: Patient experience of daily and other less frequent injection regimens in any condition (other than GHD).**   From the search conducted on 14 April 2020, seven publications were reviewed relating to patient experience switching from daily to other less frequent injection regimens in conditions other than GHD.

Three publications discussed switching from a once-daily to a three-times-weekly injection regimen in multiple sclerosis, with two reporting different data analyses conducted using the same study sample: Veneziano et al. (2017) [30] reported findings from the six-month core phase of the study, while Cutter et al. (2020) [31] reported on the six-month extension phase. Across these publications, authors reported that patients had improvements in adherence [29–31], satisfaction [31], perception of convenience [30, 31], a lower rate of AEs [30, 31], and improved QoL (mental health, fatigue) [31] with less frequent injections.

Khan et al. (2012) reported on a patient preference survey, in which 165 patients with prostate cancer were asked for their preference between hypothetical injections given once every year, six months, three months, or a month. Patients expressed an overall preference for less frequent injections, particularly for injections occurring once every six months (n = 104; 63.0%); 36 patients (21.8%) preferred injections every three months; and 25 patients (15.2%) preferred injections every 12 months. No patients preferred monthly injections [35].

Osborne et al. (2012) assessed the perceived health utilities of injections given every two weeks, once monthly, or every three months among patients with schizophrenia. Health utility scores associated with less frequent injections were found to be significantly ($p<0.001$) higher for both analysis of variance (ANOVA) and Friedman's test. The mean utility score for injections every two weeks, four weeks, and three months was 0.61 (standard deviation [SD] 0.25), 0.65 (SD 0.24), and 0.70 (SD 0.23) respectively [32].

Mathews et al. (2019) reported on patient preference for once monthly versus once every three months injections in schizophrenia. Patients expressed a preference for less frequent injections, which were associated with less injection site pain, fewer healthcare interactions, and fewer incidences of AEs. In addition, injections every three months were associated with a decrease in caregiver burden without compromising efficacy [33].

Cornford et al. (2018) compared treatment burden associated with injections every three months versus every six months in prostate cancer. After switching to injections every six months, patients reported fewer AEs and a significant ($p<0.0001$) reduction in patient-healthcare interactions [34].

## Discussion

Incorporating patients' experience in the treatment development process has become increasingly important [18]. The primary focus of this paper has therefore been on exploring and documenting the patient experience of switching from a daily to a weekly injection regimen to treat GHD; a secondary focus was on the patient experience of less frequent regimens in GHD and other conditions.

The three publications comparing switching from a daily to a weekly injection regimen in aGHD found that the weekly regimen was associated with benefits to patients such as greater convenience and higher adherence [24–26]. In addition, patients using the weekly regimen reported experiencing AEs at a lower rate (measured in events per 100 patient-years) while maintaining the same level of efficacy over the course of the studies being reported [24, 25].

The perceived benefits of a reduced injection frequency can be further supported by a recent DCE conducted with children and adolescents (with their caregivers), as well as adults,

with GHD. Here, across age groups, 75% of patients indicated they would choose a hypothetical weekly injection over their current daily injection, and the frequency of injection emerged as the most important treatment attribute for patients across age groups. These results reflect the level of burden experienced by patients who need to take daily injections, and their desire for less frequent injections while maintaining efficacy. It could reasonably be expected that maintaining efficacy while providing longer intervals between injections would improve patients' treatment experience, leading to increased treatment adherence and better outcomes over the longer term [11, 14, 28, 36, 37].

Further supporting these findings in GHD, similar results were identified in other diseases. Two publications comparing switching from a daily to a weekly injection regimen in type 2 diabetes and hypothyroidism, and four publications comparing switching from a daily to other less frequent injection regimens (e.g., three times weekly, once monthly) found that patients using less frequent regimens consistently reported increased adherence, convenience, and satisfaction.

Less frequent injections were also associated with decreased treatment burden, including fewer injection-related AEs. For patients with prostate cancer, whose injections must be administered by a healthcare provider, fewer injections meant fewer healthcare visits and less time needed for commuting to and from those visits. It should be noted, however, that this was not universally viewed as positive by patients, for whom such visits provide them an opportunity to consult with their healthcare provider and keep informed on the status of their condition.

Reduction in treatment burden was not limited only to patients. Caregivers of patients with schizophrenia reported that switching to a less frequent injection regimen decreased the burden of ensuring that patients were adherent to treatment. The caregiver impact of administering daily injections, particularly in very young children with pGHD, should not be underestimated.

As in GHD, patients with other conditions were also found to prefer a weekly or less frequent injection regimen to a daily regimen when offered the choice. Also as in GHD, less frequent injection regimens evaluated for efficacy were found to be non-inferior to daily injection regimens and, in some instances, more efficacious because of improved adherence [16, 24, 25, 29–31]. These findings are consistent with the known benefits of patients switching from a daily to a weekly regimen in oral medications, which include increased adherence, greater treatment satisfaction, and improved QoL [15, 17, 38].

Overall, the evidence presented in the reviewed publications suggests that patients with GHD switching from a daily to a weekly injection regimen are more likely to adhere to, prefer, and be satisfied with a weekly regimen, hence resulting in better outcomes.

It should be noted that this pragmatic, targeted review identified only two publications evaluating the adult patient experience of switching from a daily to a weekly injection regimen in GHD, and a third publication that evaluated patient attitudes about injection regimen in a DCE. This is not surprising, as long-acting growth hormone formulations are still a relatively recent development; most publications in GHD that were reviewed sought to evaluate efficacy or safety rather than patient experience of less frequent injections. As the development of these long-acting formulations continues, more information should become available about the effect of injection regimen on patients' treatment experience over time. Having long-term, real-world data will be important to assess the true impact of less frequent injections on efficacy, adherence, and compliance. Because compliance and adherence may change over time regardless of administration schedule, future research should examine these aspects in the context of a less frequent injection schedule.

It should also be noted that a significant proportion of individuals who currently receive growth hormone injections are children and adolescents. While it is possible that adult,

adolescent, and pediatric patients may experience some of the same burdens associated with daily injections, the two studies identified in this research discuss only the adult patient experience of switching from a daily injection regimen to a weekly injection regimen. Additional research could help to elucidate and document the specific experience of switching from a daily to a weekly injection regimen in children and adolescents with GHD, as well as their caregivers. Childhood and adolescence are formative stages in a patient's life and understanding the impact of treatment burden, and how to possibly reduce such burden could be beneficial for patients and caregivers alike.

The publications reviewed here reported on a variety of health conditions and types of treatment administration (e.g., at home or at a clinic, different injection devices). While it is possible to observe consistencies across these different treatment administrations, results presented in each publication need to be interpreted in their respective contexts. Additionally, it cannot be assumed that results experienced by patients with one disease will also be experienced by patients with another. It is to be hoped that further research will be undertaken focusing on the role that injection regimen plays in the experience of individuals who take long-term treatments and therapies that require frequent (e.g., daily) treatment regimens including GHD.

In GHD, the need for daily injections is burdensome to patients (particularly so to children and adolescents along with their caregivers) [39, 40]. Empirical evidence to date indicates that a weekly injection regimen has the potential to improve patient treatment experience, increase adherence, and over the longer term promote better health outcomes than a daily injection regimen.

## Limitations

As anticipated, this review revealed a paucity of data relating to the benefits of patients moving from a daily to a less frequent injection regimen. While a significant number of publications were identified comparing daily and weekly injection regimens in GHD and other conditions, most focused on comparing efficacy results or clinical findings; only 13 publications focused on the patient experience of switching injection regimens. While this may reflect a limitation in the search conducted (because of practical limitations, it was not possible to review all available publications related to injection regimens), the small number of publications identified may also reflect a gap in the current knowledge base about the patient experience of injection regimens. The 13 publications that were included in this analysis presented a range of benefits experienced by patients switching from daily to less frequent injections, including those switching from a daily injection regimen to a weekly injection regimen in GHD. Further, while a PICOS framework (Fig 2) was utilized to ensure that the articles selected for review met the study inclusion and exclusion criteria, the variability in therapeutic area, treatment regimen, and methodologies between each reviewed study means that there may be confounding factors not presented in this article that may influence the findings of each study. However, results from this literature review are useful given the limited amount of published data on this topic.

## Conclusions

The data presented here suggest that, in GHD and other disease areas, less frequent injection administrations are preferred by and are less burdensome to patients. Less frequent injections can positively benefit patients by improving adherence over the short-term and subsequently improving long-term clinical outcomes. Clinicians may consider available weekly regimens as an effective and preferred alternative for patients and their caregivers, in particular in pGHD where missed injections can negatively impact treatment outcomes. More research is needed

to better understand the real-world benefits of injectable therapies that require less frequent administration.

## Supporting information

**S1 Checklist. PRISMA 2009 checklist.**
(PDF)

**S1 Dataset. Minimal underlying data set.**
(DOCX)

**S1 Table. PICOS framework for systematic reviews.**
(PDF)

## Author Contributions

**Conceptualization:** Jane Loftus, Andrew Yaworsky, Diane Turner-Bowker.

**Data curation:** Megan McLafferty, Sylvia Su, Roger E. Lamoureux.

**Formal analysis:** Andrew Yaworsky, Megan McLafferty, Sylvia Su, Roger E. Lamoureux.

**Funding acquisition:** Jane Loftus, Carl L. Roland, Diane Turner-Bowker.

**Investigation:** Andrew Yaworsky, Diane Turner-Bowker, Megan McLafferty, Sylvia Su, Roger E. Lamoureux.

**Methodology:** Jane Loftus, Andrew Yaworsky, Carl L. Roland, Diane Turner-Bowker, Roger E. Lamoureux.

**Project administration:** Andrew Yaworsky, Roger E. Lamoureux.

**Supervision:** Andrew Yaworsky, Diane Turner-Bowker, Roger E. Lamoureux.

**Visualization:** Jane Loftus, Andrew Yaworsky, Carl L. Roland, Megan McLafferty, Sylvia Su, Roger E. Lamoureux.

**Writing – original draft:** Andrew Yaworsky, Diane Turner-Bowker, Roger E. Lamoureux.

**Writing – review & editing:** Jane Loftus, Andrew Yaworsky, Carl L. Roland, Diane Turner-Bowker, Megan McLafferty, Sylvia Su, Roger E. Lamoureux.

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
