## [Decision Letter · Decision Letter 0]

8 Aug 2022

PONE-D-21-40108Experience of switching from a daily to a less frequent administration of injection treatments – A review of literaturePLOS ONE

Dear Dr. Yaworsky,

Thank you for submitting your manuscript to PLOS ONE. After careful consideration, we feel that it has merit but does not fully meet PLOS ONE’s publication criteria as it currently stands. Therefore, we invite you to submit a revised version of the manuscript that addresses the points raised during the review process.

We look forward to receiving your revised manuscript.

Kind regards,

Tai-Heng Chen, M.D.

Academic Editor

PLOS ONE

Journal Requirements:

2. Thank you for stating the following in the Competing Interests section: "I have read the journal's policy and the authors of this manuscript have the following competing interests: Jane Loftus and Carl L Roland are full-time employees of Pfizer. Andrew Yaworsky, Megan McLafferty, Sylvia Su, and Roger Lamoureux are employees of Adelphi Values LLC, who were paid consultants to Pfizer in connection with this research and development of this manuscript. Diane Turner-Bowker was an employee of Adelphi Values LLC at the time that the research was conducted."

We note that you received funding from a commercial source: [Name of Company]

Reviewers' comments:

Reviewer's Responses to Questions

**Comments to the Author**

1. Is the manuscript technically sound, and do the data support the conclusions?

Reviewer #1: Yes

Reviewer #2: Partly

2. Has the statistical analysis been performed appropriately and rigorously? 

Reviewer #1: Yes

Reviewer #2: N/A

3. Have the authors made all data underlying the findings in their manuscript fully available?

Reviewer #1: Yes

Reviewer #2: Yes

4. Is the manuscript presented in an intelligible fashion and written in standard English?

Reviewer #1: Yes

Reviewer #2: Yes

5. Review Comments to the Author

Reviewer #1: This is an interesting systematic review of the potential benefits patients can experience when switching from a daily to weekly injection regimen for GHD or, in any type of transition to less frequent dosing, for any other disease.

This is an important topic because, as the scant studies in literature show, it is rarely explored by clinicians.

The authors show that compliance with the less frequent schedule is greater while efficacy is comparable and that this is documented in both GHD and other diseases.

It is important to underline that childhood and adolescence are phases of possible fragility, in particular when it comes to illnesses, and that the impact of a greater appreciation of the method of administration could be reflected in multiple areas (psychological ... relational effects ..)

Small criticisms

Data for GHD are obtained from trials; we all know that the results of real medical practice are different from those of a study program. It should be considered that, after years of treatment, compliance could change regardless of the administration schedule.

In the discussion session only one comment should be added by the authors

Reviewer #2: An interesting and potentially useful contribution to the important field of patient experience in long term clinical care. Issues for clarification:

1. It's unclear why the review focuses on the benefits of transition from daily to weekly GH regimens - the full range of patient experience seems more relevant.

2. The search strategy included two subsidiary searches for non-GH treatments. No information is provided on these search strategies - this is required before any judgement can be made on appropriateness. This is an important issue given the very wide diversity of possible inclusions / exclusions.

3. The numbers of papers identified in the different search blocks isn't clear, particularly in 'Search 2'. Please clarify.

4. The key challenge in this study is the absence of any quality measures in relation to the 13 studies reported. The diversity of methods, goals, illnesses and treatment modalities across such a small number of studies raises significant potential for methodological problems and the introduction of confounders, limiting comparability. Very little information is available on these issues in Table 2 and no standardised quality assessment tool is proposed. Please comment.

5. Without a clear quality assessment and commentary, this isn't a normal systematic review and might best be presented as a pragmatic review. However, the assessment of quality is important given the challenges mentioned above and the utility of the study is limited without such insight.

6. PLOS authors have the option to publish the peer review history of their article (what does this mean?). If published, this will include your full peer review and any attached files.

Reviewer #1: No

Reviewer #2: **Yes: **Gerard Bury

---

## [Author Response · Author response to Decision Letter 0]

18 Oct 2022

To whom it may concern:

Thank you again for the feedback provided by the academic editor and reviewers for our manuscript entitled “Experience of switching from a daily to a less frequent administration of injection treatments” (PONE-D-21-40108).

We have reviewed the feedback from both the editor and reviewers, and our responses to the feedback related directly to the content of the manuscript and the associated tables, and figures are summarized in a table included in the "Response to Reviewers" document uploaded. In-text edits have been made within the manuscript in track-changes and, where appropriate, we have included reference to the line numbers where these edits have been made within the response table included in the attached document that summarizes our responses to the reviewer feedback. We do hope that the revisions to the manuscript address the concerns of the reviewers, but if there is any further feedback, we are happy to revise further as needed. Please let us know.

One ask that related to supporting information associated with the manuscript was for the co-authors to revise our competing interest disclosures to specify who funded the study – and to indicate that these competing interests do not alter our adherence to the PLOS ONE policies on data sharing. This revised statement is included in the "Response to Reviewers" letter that has been uploaded, as well as included in the manuscript itself.

Another point of consideration that was made by the editor following review was a request for our “minimal data set.” Based on the context of the study (i.e., a literature review) there is no traditional “data set.” However, we do provide the full list of articles, along with citations, as part of Table 2 should anybody wish to re-create our analysis. We have uploaded the list of articles as a minimal data set should the journal wish to publish this, though we do feel that Table 2 and the associated citation list should be sufficient. 

As noted above, we hope the changes to the manuscript and the additional information provided improve the manuscript and allow it to be fit for publication in PLOS ONE. If there are any further questions or feedback surrounding this manuscript, please let us know!

Thank you again for the opportunity and for considering this manuscript for publication.

Best regards,

Andrew Yaworsky

andrew.yaworsky@adelphivalues.com | +1 (617) 720-0001

---

## [Decision Letter · Decision Letter 1]

15 Nov 2022

Experience of switching from a daily to a less frequent administration of injection treatments – A review of literature

PONE-D-21-40108R1

Dear Dr. Yaworsky,

We’re pleased to inform you that your manuscript has been judged scientifically suitable for publication and will be formally accepted for publication once it meets all outstanding technical requirements.

Kind regards,

Tai-Heng Chen, M.D.

Academic Editor

PLOS ONE

Reviewers' comments:

Reviewer's Responses to Questions

**Comments to the Author**

1. If the authors have adequately addressed your comments raised in a previous round of review and you feel that this manuscript is now acceptable for publication, you may indicate that here to bypass the “Comments to the Author” section, enter your conflict of interest statement in the “Confidential to Editor” section, and submit your "Accept" recommendation.

Reviewer #1: All comments have been addressed

Reviewer #2: All comments have been addressed

2. Is the manuscript technically sound, and do the data support the conclusions?

Reviewer #1: Yes

Reviewer #2: Yes

3. Has the statistical analysis been performed appropriately and rigorously? 

Reviewer #1: Yes

Reviewer #2: N/A

4. Have the authors made all data underlying the findings in their manuscript fully available?

Reviewer #1: Yes

Reviewer #2: Yes

5. Is the manuscript presented in an intelligible fashion and written in standard English?

Reviewer #1: Yes

Reviewer #2: Yes

6. Review Comments to the Author

Reviewer #1: The Author accepted my suggestions and added them in the discussion section

The paper, is now suitable for publication on PLOS One

Reviewer #2: Thank you for addressing the issues raised. The paper should now be considered for publication by the editor, taking into account the information provided on funding.

7. PLOS authors have the option to publish the peer review history of their article (what does this mean?). If published, this will include your full peer review and any attached files.

Reviewer #1: **Yes: **Silvia Grottoli

Reviewer #2: **Yes: **Gerard Bury

---

## [Editor Report · Acceptance letter]

18 Nov 2022

PONE-D-21-40108R1 

Experience of switching from a daily to a less frequent administration of injection treatments 

Dear Dr. Yaworsky:

I'm pleased to inform you that your manuscript has been deemed suitable for publication in PLOS ONE. Congratulations! Your manuscript is now with our production department. 

Kind regards, 

on behalf of

Dr. Tai-Heng Chen 

Academic Editor

PLOS ONE